# External validation and recalibration of an incidental meningioma prognostic model – IMPACT: protocol for an international multicentre retrospective cohort study

Abdurrahman I Islim [1,2] Christopher P Millward [1,2] Rory J Piper,[3] Daniel M Fountain,[4] Shaveta Mehta,[1,5] Ruwanthi Kolamunnage-Dona [6] Usama Ali,[7] Shelli Diane Koszdin,[8] Theo Georgious,[9] Samantha J Mills,[10] Andrew R Brodbelt,[1,2] Ryan K Mathew [11,12] Thomas Santarius,[13,14] Michael D Jenkinson,[1,2] on behalf of the IMPACT Study Investigators, International Consortium on Meningioma (ICOM) and British Neurosurgical Trainee Research Collaborative (BNTRC)

For numbered affiliations see end of article.

**Correspondence to**
Dr Michael D Jenkinson;
Michael.Jenkinson@liverpool.ac.uk and
Dr Abdurrahman I Islim;
a.islim@doctors.org.uk

## ABSTRACT

**Introduction** Due to the increased use of CT and MRI, the prevalence of incidental findings on brain scans is increasing. Meningioma, the most common primary brain tumour, is a frequently encountered incidental finding, with an estimated prevalence of 3/1000. The management of incidental meningioma varies widely with active clinical-radiological monitoring being the most accepted method by clinicians. Duration of monitoring and time intervals for assessment, however, are not well defined. To this end, we have recently developed a statistical model of progression risk based on single-centre retrospective data. The model Incidental Meningioma: Prognostic Analysis Using Patient Comorbidity and MRI Tests (IMPACT) employs baseline clinical and imaging features to categorise the patient with an incidental meningioma into one of three risk groups: low, medium and high risk with a proposed active monitoring strategy based on the risk and temporal trajectory of progression, accounting for actuarial life expectancy. The primary aim of this study is to assess the external validity of this model.

**Methods and analysis** IMPACT is a retrospective multicentre study which will aim to include 1500 patients with an incidental intracranial meningioma, powered to detect a 10% progression risk. Adult patients ≥16 years diagnosed with an incidental meningioma between 1 January 2009 and 31 December 2010 will be included. Clinical and radiological data will be collected longitudinally until the patient reaches one of the study endpoints: intervention (surgery, stereotactic radiosurgery or fractionated radiotherapy), mortality or last date of follow-up. Data will be uploaded to an online Research Electronic Data Capture database with no unique identifiers. External validity of IMPACT will be tested using established statistical methods.

**Ethics and dissemination** Local institutional approval at each participating centre will be required. Results of the study will be reported through peer-reviewed articles and

### Strengths and limitations of this study

► The first multicentre international study to investigate the prognosis of incidental intracranial meningiomas.
► The study will include a large cohort of 1500 patients.
► The longitudinal study design with serial collection of clinical and imaging data will provide a unique insight into meningioma behaviour and provide a platform for future investigation of novel biomarkers.
► The retrospective nature of the study may bias patient and information selection.
► Results of the study may be biased by clinician and patient management preference in each participating centre.

conferences and disseminated to participating centres, patients and the public using social media.

## INTRODUCTION

Meningiomas have the highest incidence rate among all primary central nervous system tumours. Descriptive studies from Europe and North America suggest this rate is between 4.20 and 8.58 per 100 000 individuals.[1] [2] Wider access and increased use of MRI and CT have led to a marked rise in the number of incidental findings in clinical and research settings. Meningiomas comprise 15% of incidental findings on brain MRI and have a prevalence of 3 per 1000.[3] A recent study of the Surveillance, Epidemiology, and End Results database demonstrated

a substantial increase in the detection of smaller, incidental tumours; between 2004 and 2012, the proportion of meningiomas<1 cm in diameter, diagnosed in a given year, increased in a linear fashion from 6% to 11%.[4] Incidental, asymptomatic meningiomas cause patient anxiety and uncertainly around the need for future treatment and often prompt clinicians to commence long-term MRI and clinical follow-up. International consensus guidelines by the European Association of Neuro-Oncology and National Comprehensive Cancer Network suggest active monitoring with MRI as first line for managing these tumours,[5 6] but data to advise on the optimal follow-up duration and screening intervals are currently lacking.[7]

Previous studies have identified prognostic radiological factors that are associated with the risk of meningioma growth and development of clinical symptoms; yet the timing of such progression is poorly defined.[8-10] Moreover, clinical factors such as patient comorbidity and performance status remain unexplored in relation to prognosis but are highly relevant. The patient with an incidental meningioma wants to know whether their tumour will grow and become symptomatic such that it will require safe treatment within their healthy lifetime.

To this end, a recent retrospective cohort study of incidental meningioma patients in the UK was conducted to assess the utility of combining routinely available radiological and clinical factors to develop a prognostic model for the risk of incidental meningioma progression during active monitoring.[11] The model Incidental Meningioma: Prognostic Analysis Using Patient Comorbidity and MRI Tests (IMPACT) could be used as a tool to guide active monitoring strategies for patients with an incidental asymptomatic meningioma within the first 10 years of diagnosis, however, validation with external datasets is required.

The primary aim of this international retrospective cohort study of incidental meningioma is to externally validate and calibrate the prognostic model IMPACT, accessible using https://www.impact-meningioma.com. These data will provide insight into the incidence, epidemiology, presentation, management and long-term outcomes of incidental meningioma, which will inform the development of clinical guidelines and identify areas for future research.

## THE IMPACT MODEL
The model, based on MRI parameters, stratifies patients with an incidental meningioma into three risk groups: low, medium and high risk. These MRI parameters are as follows: meningioma volume, meningioma hyperintensity, peritumoural signal change and proximity to critical neurovascular structures. The predictive function was built using an internally validated Cox regression model. Patients were also stratified in the model based on age, comorbidity and performance status using competing risk analyses.

## OBJECTIVES
### Primary objective
To externally validate the prognostic model IMPACT.

### Secondary objectives
► To update the parameters of the prognostic model IMPACT if measures of external validation demonstrate a poor fit, and internally validate the updated model.
► To determine the growth patterns of incidental meningiomas.
► To examine the MRI and pathology features of meningiomas subject to surgical resection.
► To determine the risk of post-intervention complications and tumour recurrence/growth for meningiomas subject to surgery, stereotactic radiosurgery (SRS) or fractionated radiotherapy (fRT).
► Assess the economic implications of stratifying follow-up according to risk of disease progression.

## METHODS AND ANALYSIS
### Study design
This will be a retrospective, international multicentre cohort study. The study will include incidental meningioma patients managed at each participating centre. Cases will be identified by the local site research teams using existing patient medical records. Baseline clinical and radiological characteristics, tumour management, and clinical and radiological outcomes will be collected and recorded (anonymised data) on a secure database by the local investigators. Since this study falls within the remit of clinical outcomes audit, individual patient consent is not required. The study will collect data from the medical records for patients newly-diagnosed over a 2-year period between 1 January 2009 and 31 December 2010. This is an observational study and will not alter routine patient care.

### Study population and eligibility criteria
The study will include adults (≥16 years of age) with a newly identified incidental intracranial meningioma, as per radiology report, diagnosed between 1 January 2009 and 31 December 2010. Radiological diagnosis is expected to be based on the presence of an extra-axial lesion with broad-based attachment along the dura showing contrast enhancement. The accepted definition of an incidental finding is 'a previously undetected abnormality of potential clinical relevance that is unexpectedly discovered and unrelated to the purpose of the examination'.

Exclusion criteria are as follows:
► History of cranial radiation >5 years before diagnosis.
► History of neurofibromatosis type 2.
► Surgical resection which revealed a different histopathological diagnosis.
► Unavailability of medical notes.

### Patient identification
Eligible patients can be identified using local radiology information systems, for example the Computerised

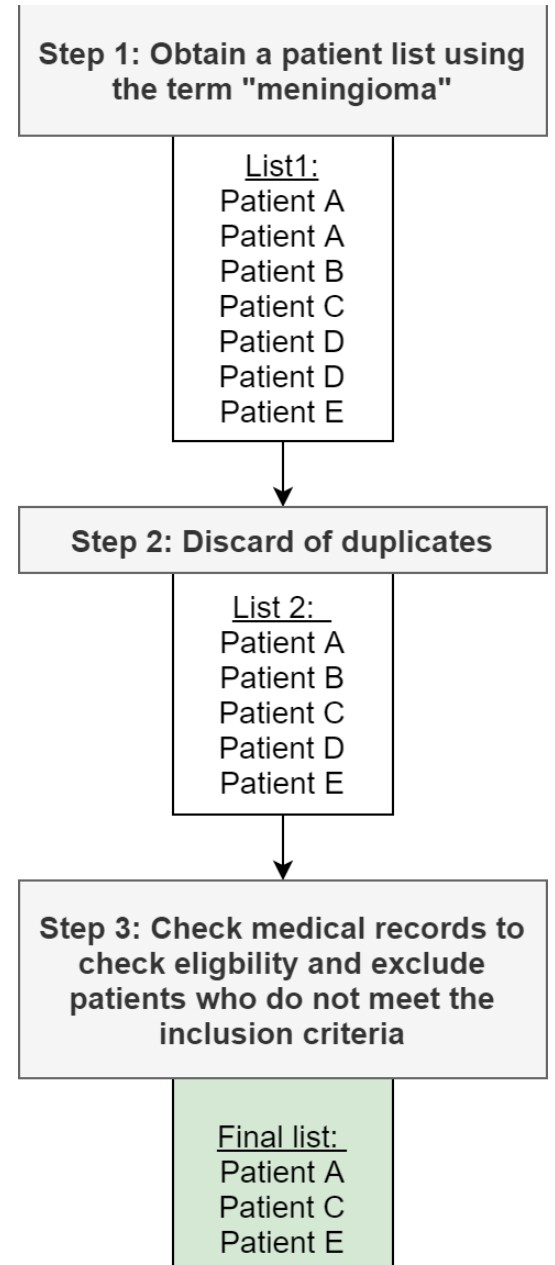

**Step 1: Obtain a patient list using the term "meningioma"**

List1:
Patient A
Patient A
Patient B
Patient C
Patient D
Patient D
Patient E

**Step 2: Discard of duplicates**

List 2:
Patient A
Patient B
Patient C
Patient D
Patient E

**Step 3: Check medical records to check eligbility and exclude patients who do not meet the inclusion criteria**

Final list:
Patient A
Patient C
Patient E

**Figure 1** Process of creating a patient list at each study site.

Radiological Information System tool. The search strategy will involve review of the medical records of all patients managed with a meningioma at participating centres and exclusion of those that do not meet the selection criteria (figure 1).

### Sample size

For external validation studies, a minimum of 100 events is required.[12] The risk of incidental meningioma progression is estimated to be 10%.[11] Based on this, data for 1000 patients will be required. To account for variability in the progression risk, follow-up regimens and loss to follow-up, we will aim to include a minimum of 1500 patients across participating centres. An interim analysis will be conducted after data for 500 patients have been collected to assess for the risk of incidental meningioma progression and review the required number of patients.

### Study endpoints

#### Primary endpoint

Disease progression will be defined using a composite endpoint comprising of new symptom development, meningioma-specific mortality, meningioma growth (absolute growth rate (AGR) $\geq 2$ cm$^3$/year or AGR $\geq 1$ cm$^3$/year+relative growth rate (RGR) $\geq 30\%$ /year), development or increase of peritumoural brain oedema (defined as increased signal intensity on T2/fluid attenuated inversion recovery (FLAIR) MRI), venous sinus invasion and meningioma volume exceeding 10 cm$^3$. The first two criteria denote clinical progression while the latter three are related to loss of window of curability. Venous sinus invasion and peritumoural oedema can prevent complete surgical resection.[13 14] Peritumoural oedema and a meningioma volume >10 cm$^3$ are relative contraindications to SRS.[15 16]

#### Secondary endpoints

Intervention (surgery, SRS or ƒRT) and mortality unrelated to the meningioma.

### Data collection

Data will be collected at each centre by members of the local team. Data will be collected from the patient's medical and radiology records. All clinical and radiological information collected for this study by the local investigators should be available routinely and no extra patient assessment will be required. Data will be collected and stored online through a secure University of Liverpool server running the Research Electronic Data Capture (REDCap) web application and using the patient unique study number. Local investigators will be given secure REDCap project server login details. No patient identifiable information will be uploaded or stored on the REDCap database. The study number (site ID_patient ID) is generated by REDCap on creating a new patient record in the database. The clinical team can only view the records of patients from their own centre. All local investigators will store a copy of the link between the patient's unique study number and their patient identifiers on a secure password protected computer, using a blank link file provided by the study team.

### REDCap database

REDCap is a secure web application for building and managing online databases. Access to REDCap will be provided by the Liverpool Clinical Trials Centre, University of Liverpool, a partner of the REDCap consortium. Database programmers will oversee the development of a data collection tool (online supplemental appendix 1) which can be accessed using any electronic device with internet access. The database will be built to comply with the UK's Data Protection Act 2018 and the European Union's General Data Protection Regulation (GDPR).

**Table 1** WHO performance status classification and the age-adjusted Charlson Comorbidity Index

| WHO performance status classification | | Age-adjusted Charlson Comorbidity Index | | |
|---|---|---|---|---|
| Score | Description | Condition | | Weight |
| 0 | Able to carry out all normal activity without restriction | Age (years) | <50 | 0 |
| 1 | Restricted in strenuous activity but ambulatory and able to carry out light work | | 50–59 | 1 |
| 2 | Ambulatory and capable of all self-care but unable to carry out any work activities; up and about more than 50% of waking hours | | 60–69 | 2 |
| 3 | Symptomatic and in a chair or in bed for greater than 50% of the day but not bedridden | | 70–79 | 3 |
| 4 | Completely disabled; cannot carry out any self-care; totally confined to bed or chair. | | ≥80 | 4 |
| 5 | Dead | Myocardial infarction | | 1 |
| | | Congestive heart failure | | 1 |
| | | Peripheral vascular disease | | 1 |
| | | Hemiplegia | | 2 |
| | | Cerebrovascular disease | | 1 |
| | | Pulmonary disease | | 1 |
| | | Diabetes | | 1 |
| | | | With end-organ damage | 2 |
| | | Renal disease | | 2 |
| | | Liver disease | Mild | 1 |
| | | | Severe | 3 |
| | | Peptic ulcer disease | | 1 |
| | | Cancer | | 2 |
| | | | Metastatic | 6 |
| | | Dementia | | 1 |
| | | Connective tissue disease | | 1 |
| | | AIDS | | 6 |
| | | Hypertension | | 1 |
| | | Skin ulcers/cellulitis | | 2 |
| | | Depression | | 1 |
| | | On Warfarin | | 1 |

Quality assessment of the tool will be done over two phases. Phase 1 will involve local testing of the tool using pre-existing data.[11] Phase 2 will expand testing to three to five additional participating centres. After completion of phase 2, the data collection tool will be made live for use by the participating sites.

### Recorded variables
#### Baseline clinical variables
Age at diagnosis, sex, ethnicity, World Health Organisation (WHO) performance status (PS) and the age adjusted Charlson Comorbidity Index (ACCI) (table 1).[17–19] These factors will only be recorded at baseline.

#### Baseline radiological variables
Baseline imaging variables assessed will be:
► Single or multiple intracranial meningioma.

► Tumour signal intensity compared with the contralateral grey matter on FLAIR and T2-weighted (T2) MRI (hypo/iso/hyper) (figure 2).
► Peritumoural signal intensity in relation to tumour volume using the signal change present on FLAIR and T2 MRI (0%–5%/6%–33%/34%–66%/67%–100%; adapted from the Visually AcceSAble Rembrandt Images MR features for gliomas[20]).
► Meningioma volume using the ABC/2 formula on contrast-enhanced T1-weighted MRI/CT: (A) maximum meningioma diameter on axial plane, (B) diameter perpendicular to (A) and (C) maximum height on coronal/sagittal plane, not taking into the account the dural tail.
► Meningioma location classed into non-skull base and skull base and further subcategorised according to the International Consortium on Meningioma classification system (online supplemental appendix 2).

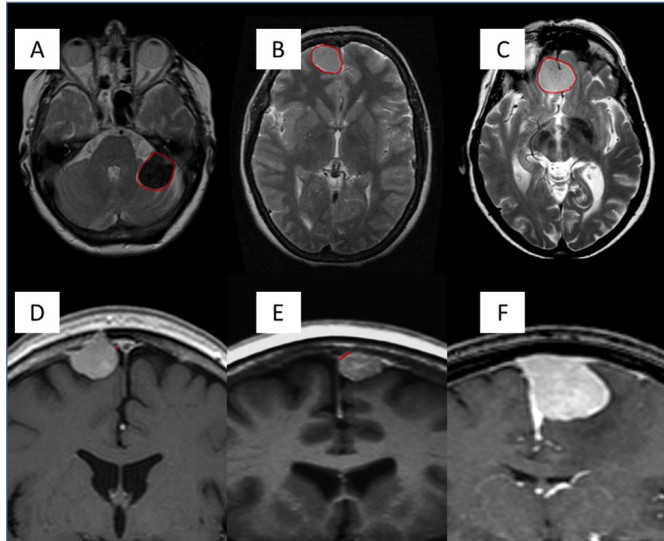

**Figure 2** (A–C) T2 MR axial sequences showing the three levels of tumour intensity (circle). (A) Hypointense. (B) Isointense. (C) Hyperintense. (D–F) T1-weighted MR with gadolinium (contrast) showing the relationship between the meningioma and the nearby venous sinus (SSS). (D) Separate as there is no clear attachment to the sinus wall. (E) In direct contact with the lateral wall of the sinus. (F) Clear macroscopic distortion and invasion of the sinus. SSS, superior sagittal sinus.

▶ Proximity to major dural venous sinuses (superior sagittal sinus/transverse sinus/sigmoid sinus/cavernous sinus/the confluence of sinuses) categorised as separate (within 10 mm), in direct contact with its wall, or invading, excluding the dural tail (figure 2).

▶ Contact with critical neurovascular structures (ie, internal carotid artery and optic apparatus).

Meningiomas that fulfil one of the two previous categories are said to be in proximity to critical neurovascular structures. A video manual prepared by the study team will be made available to assist with standardisation and quality assurance of scan interpretation across participating centres.

### Management strategy

Management strategies will include active monitoring, intervention (surgery, SRS and ʄRT) or discharge from outpatient clinic care (figure 3). Active monitoring is defined as regular surveillance imaging and outpatient clinical observation. Recorded factors will include:

▶ Number of scans, and interval between them (months).

▶ For each scan: peritumoural signal intensity, venous sinus involvement and meningioma volume.

▶ Each scan will be examined alongside its corresponding outpatient clinic appointment for any evidence of meningioma-related symptoms (motor/sensory/language/cognitive/seizure/headache/other).

▶ The outcome of each clinical encounter (ie, outpatient appointment) will be recorded (resume follow-up/surgery/SRS/ʄRT/hospital discharge).

Intervention details; if performed, will also be recorded. These will include indication for intervention (clinical-radiological/clinical/radiological/patient preference) and time to intervention. For patients treated with clinical-radiological or clinical progression, status of meningioma-related neurological morbidity will be noted.

For surgery, the following will additionally be recorded:

▶ Simpson grade (as recorded by the surgeon in the operative notes).[21]

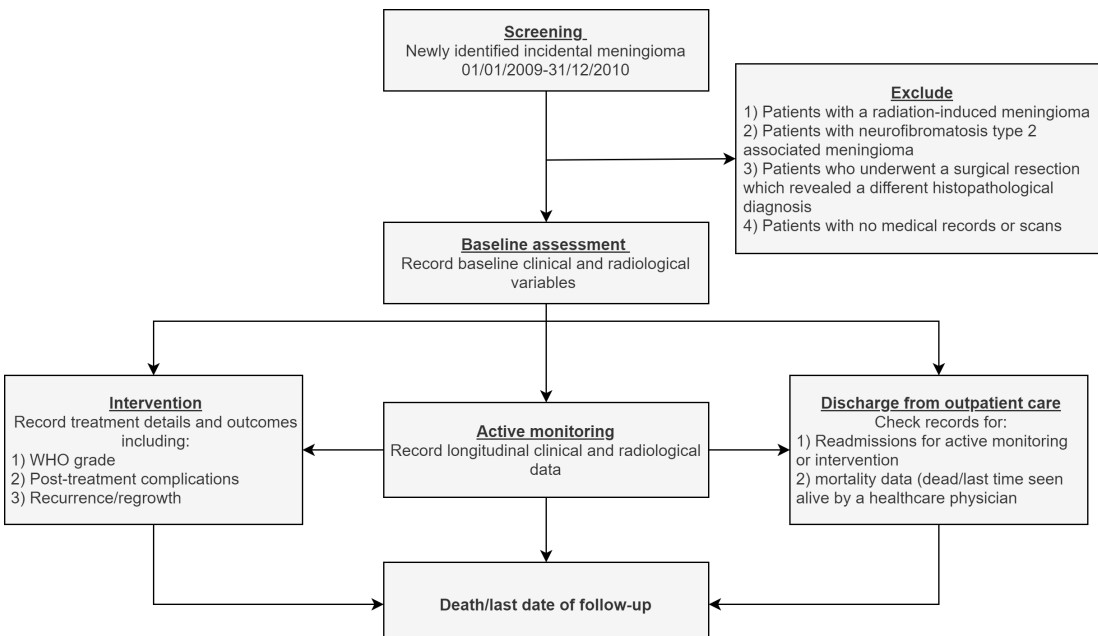

**Figure 3** Study flow chart depicting the process of patient identification and possible management options within the study.

| Table 2 | Landriel-Ibañez classification of neurosurgical complications |
|---|---|
| Grade 1 | Any non-life-threatening deviation from normal postoperative course, not requiring invasive treatment |
| Grade 1a | Complication requiring no drug treatment |
| Grade 1b | Complication requiring drug treatment |
| Grade 2 | Complication requiring invasive treatment such as surgical, endoscopic, or endovascular interventions |
| Grade 2a | Complication requiring intervention without general anaesthesia |
| Grade 2b | Complication requiring intervention with general anaesthesia |
| Grade 3 | Life-threatening complications requiring management in an intensive care unit |
| Grade 3a | Complication involving single organ failure |
| Grade 3b | Complication involving multiple organ failure |
| Grade 4 | Complication resulting in death |
| Surgical Complications | Adverse events that are directly related to surgery or surgical technique |
| Medical Complications | Adverse events that are not directly related to surgery or surgical technique |
| Suffix 'T' (Transient) | New neurologic deficit improving within 30 days of surgical procedure; can be added to each grade of complication |
| Suffix 'P' (Persistent) | New neurologic deficit extending beyond 30 days of surgical procedure; can be added to each grade of complication |

► WHO grade (classified according to the WHO system in use at the time of surgery and updated according to the WHO 2016 classification dependent on pathologists' availability[22]) and presence of any reported brain invasion (yes/no/not reported).

► Postoperative medical and surgical complications recorded at 30 days (Landriel-Ibañez classification (table 2).[23]

► Follow-up duration (months).

► WHO PS preoperatively and postoperatively and at the last follow-up appointment.

► Recurrence on contrast-enhanced MRI during that time (yes/no) and if recurred then the time to recurrence.

For SRS and fRT:

► Fractionated dose (fRT), number of fractions (fRT) and total dose (fRT /SRS).

► Early and late (≥3 months) toxicity (assessed by CTCAE V.5.0, https://ctep.cancer.gov/protocoldevelopment/electronic_applications/ctc.htm).

► Duration of follow-up postradiation (months).

► WHO PS preradiation and post-radiation and at the last follow-up appointment.

► Progression/regrowth on contrast-enhanced MRI during that time (yes/no) and if progressed/regrew then the time to progression/regrowth.

For patients discharged from outpatient care, data sources will be checked for any readmissions/rescans thought to be attributed to the incidental meningioma within the study time frame; date of diagnosis up to the date of data entry. Outcome following readmissions/rescans will be noted.

## Overall outcomes

Overall outcomes by the end of the study period (discharge from outpatient care/lost to follow-up/dead/under on-going active follow-up) and follow-up durations will be recorded. Any deaths encountered during follow-up will

be recorded. The medical records for patients who are discharged will also be examined for mortality data.

## Data quality assurance

An e-learning module will be prepared by the study team. This will contain the data collection guides and video manual. On completion, each study investigator will need to undergo a five-item assessment. An iterative process in which investigators have to redo the assessment or module will dictate their progress as follows:

► An assessment percentage of 100% will indicate successful completion of the module, which will allow the investigator to collect data for the study.

► An assessment percentage less 100% will require repeating the assessment.

► Five attempts will be allowed.

► Subsequent failed attempts will entail review of the module components again.

## Planned statistical analysis

Demographic differences across groups will be explored with the $\chi^2$ test for categorical variable and the Mann-Whitney U test or Student's t-test for continuous variables. Correlation between baseline variables will be evaluated using the Pearson correlation coefficient. Normally distributed continuous variables will be expressed as mean (SD) whereas skewed variables as median (IQR). Differences will be considered statistically significant at $p < 0.05$.

## External validation

Using IMPACT, the 5-year and 10-year estimated risk of disease progression for every patient included in this cohort study will be calculated. Kaplan-Meier method will be used to obtain the observed risks. The predictive performance of IMPACT will be assessed by examining measures of calibration and discrimination. Calibration refers to how closely the predicted 5-year and 10-year risk of progression agrees with the observed risk. A calibration

plot compares the observed and predicted rates of events for each group. A perfect match indicates accurate calibration. The Brier score for censored survival data will also be calculated, which is a measure of accuracy and is the average squared deviation between predicted and observed risk; a lower score represents greater accuracy. Discrimination is the ability of the risk score to differentiate between those patients who do and those who do not experience disease progression during the study time frame. This measure is quantified by calculating the C-statistic, D-statistic and Chambless and Diao's time-dependent area under the receiver operating characteristic curve which are tailored towards censored survival data. The proportional hazards assumption of the model will be tested by examination of Schoenfeld residuals, and influential observations will be examined using DFBETA panels.

The two competing risk analyses performed to build the IMPACT model will be repeated with the external dataset and plots of cumulative incidence rate (CIR) and 95% CIs will be compared with the original cohort.[11] Patients will be split based on WHO PS into two categories: 0–1 and 2–4 and stratified by ACCI (table 1) into three groups: 0–2, 3–5 and ≥6. The first analysis will assess the CIR of primary intervention at different time points following diagnosis stratified by PS and ACCI groups and the second analysis will evaluate the CIR of disease progression. The competing event for the former will be non-meningioma-specific mortality either observed during follow-up or after being discharged from outpatient care. Patients who remain under follow-up will be censored at the last outpatient clinic appointment. Patients discharged alive from outpatient care will censored at the last time they were seen by a healthcare physician up to the date of data entry. For the disease progression analysis, four events will be considered competing in nature, discharge from outpatient care, loss to follow-up, death during follow-up or an intervention before disease progression occurred with the first three grouped together. Censoring will only be done for patients who remain under follow-up at the last clinic appointment. To test the equality across CIR groups, the Fine and Gray test will be carried out.

### Model recalibration

If calibration and discrimination measures of external validation demonstrate a poor fit, the model will be recalibrated and adjusted using the data of included patients. This will be done over four stages:

▶ Stage 1: The regression coefficients will be recalibrated. This will be done using a Cox regression model fitted with the linear predictor as the only covariate.

▶ Stage 2: The recalibrated model predictors will each be removed in a stepwise manner by a non-automated criterion-based procedure starting with the variable with a HR closest to 1. After removal of this variable, the aforementioned measures of discrimination of calibration and discrimination will be reassessed to detect model improvement. If the performance of the

model is unimproved or worsens, the variable will be reintroduced to the model. This step will be repeated in a staged manner until no further improvements are detected. Introduction of new predictive variables will be possible.

▶ Stage 3: The internal validity of the updated model will be assessed using a bootstrapping method.

▶ Stage 4: Adjusted stratification by ACCI and PS (table 1) will be performed to achieve statistically significant differences in equality across CIR groups, judged by the Fine and Gray test.

### Additional analyses

We envisage that imaging protocols in the participating centres are varied and non-standardised and thus, the growth rate for each meningioma will be determined using a joint longitudinal and event-time outcomes model which does not require regularly spaced time points, and adjusts for informative follow-up, assuming a different intercept and slope for each meningioma.[24 25] The sum of the regression coefficients of random and fixed effects for the slope estimated from the linear model will best represent the average growth rate for each meningioma. AGR will be defined as the increase in volume per year in $cm^3$ whereas RGR will be defined as the percentage increase in volume per year.

This statistical analysis plan will be reviewed prior to the final analysis of the study.

### Health economic analysis

The health economic analysis will adopt the perspective of the National Health Service in the UK. Costs related to clinic appointments and MRI scans will be calculated for the study cohort's retrospectively performed follow-up plans and compared against two follow-up regimens:

▶ The follow-up regimen proposed by the National Institute for Healthcare Excellence of 2 scans at 12 months and 5 years.

▶ The follow-up regimen using the IMPACT model, stratified by risk of progression.

### Patient and public involvement

Two patient research partners (PRPs) have been involved in the design of the study and are members of the steering committee. With the support of the Brain Tumour Charity, represented by a member of the steering committee, the aim is to engage with more PRPs and to build a partnership to be continued throughout delivery of the study and dissemination and presentation of results.

## ETHICS AND DISSEMINATION
### Study registration

It will be the responsibility of the research team at each unit to register the study as a clinical audit with their hospital's audit department in the UK, including Caldicott guardian or Information Governance approval as

required. Overseas sites will register with according to their local institutional policy.

## Local investigator responsibilities

The investigator will be responsible for the overall conduct of the study at the site and compliance with the protocol. Responsibilities may be delegated to an appropriate member of the local research team. The investigator must also be familiar with the protocol and the study requirements and it is their responsibility to ensure that all staff assisting with the study are adequately informed about the protocol and the study requirement. The principal investigator at each centre is responsible for the quality of the data recorded in the database.

## Confidentiality and data protection

No patient identifiable information will be uploaded or stored on the REDCap database. The clinical team can only view the records of patients from their own centre. All records must be identified in a manner designed to maintain patient confidentiality and must be kept in a secure storage area with limited access; all local investigators will store a copy of the link between the patient's unique study number and their patient identifiers on a secure password-protected computer, using a blank link file provided by the study team. The investigator and local research terms involved with this study may not disclose or use for any purpose other than performance of the study, any data, record, or other unpublished, confidential information. They also must comply with the requirements of the Data Protection Act 2018 and GDPR with regard to the collection, storage, processing and disclosure of personal information. Access to collated patient data will be restricted to individuals from the research team and representatives of the sponsor. Computers used to collate the data will have limited access measures via usernames and passwords. Published results will not contain any personal data that could allow identification of individual patients.

## Ownership

Ownership of the complete dataset arising from this study resides with the steering committee (named authors in this protocol). Local data collected as part of this study belongs to the local team collecting that data. However, individual clinicians must not submit any part of their individual data for publication or presentation without prior consent from the study research team. Individual participant data, after deidentification, will be made available to researchers whose proposed use of the data is approved by the original study investigators. Proposals should be directed to the primary investigator.

## Dissemination of results

The study results will be reported using the Transparent Reporting of a multivariable prediction model for Individual Prognosis Or Diagnosis checklist. The results of this study will be presented at national and international meetings and will be submitted for publication in peer-reviewed journals.

## Authorship eligibility

The list of named authors will resemble this protocol's authorship. The contribution of all investigators captured via the REDCap database, will be recognised with PubMed Citable collaborator status authorship under the umbrella of the IMPACT study investigators.

**Author affiliations**
[1]Institute of Systems, Molecular & Integrative Biology, University of Liverpool, Liverpool, UK
[2]Department of Neurosurgery, The Walton Centre NHS Foundation Trust, Liverpool, UK
[3]Department of Neurosurgery, John Radcliffe Hospital, Oxford University Hospitals NHS Foundation Trust, Oxford, UK
[4]Manchester Centre for Clinical Neurosciences, Salford Royal NHS Foundation Trust, Salford, UK
[5]Department of Clinical Oncology, The Clatterbridge Cancer Centre NHS Foundation Trust, Wirral, UK
[6]Department of Health Data Science, Institute of Population Health, University of Liverpool, Liverpool, UK
[7]Clinical Trial Service Unit and Epidemiological Studies Unit, Nuffield Department of Population Health, University of Oxford, Oxford, UK
[8]Veterans Affairs Healthcare System, Palo Alto, California, USA
[9]Brain Tumor Charity, Fleet, UK
[10]Department of Neuroradiology, The Walton Centre NHS Foundation Trust, Liverpool, UK
[11]Leeds Institute of Medical Research at St James's, University of Leeds School of Medicine, Leeds, UK
[12]Department of Neurosurgery, Leeds Teaching Hospitals NHS Trust, Leeds, UK
[13]Department of Neurosurgery, Addenbrooke's Hospital, Cambridge, UK
[14]Division of Neurosurgery, University of Cambridge School of Clinical Medicine, Cambridge, UK

**Collaborators** IMPACT Study investigators: Mohamed Abdelsadg, Suhaib Abualsaud, Amro Abuleil, Kevin Agyemang, Hanan Akbari, Likhith Alakandy, Erminia Albanese, Clarissa Alfonso, Usama Ali, Arousa Ali, Tamara Ali, Michael Amoo, Mohamed A. R. Arbab, Mutiu Asha, Kareem Austin, Khaled Badran, Jarnail Bal, Chris Barrett, Parameswaran Bhattathiri, Paul M. Brennan, Andrew R. Brodbelt, Jennifer Brown, Daniel Brown, Ferran Brugada-Bellsolà, Placido Bruzzaniti, Annabel Butcher, Rory S. Cairns, Michael Canty, Michael Cearns, Sachiv Chakravarti, Rebecca Chave-Cox, Yasir Chowdhury, Anna Craig-McQuade, Giles Critchley, Peter Crossley, Elizabeth Culpin, Alessia D'Amico, Bassam Dabbous, Pedro David Delgado-López, Mohamed Draz, Katharine J. Drummond, Rusiru T. Ekanayaka, Muhammed Elhadi, Ibrahim Elmaadawi, Omar Elmandouh, Mazin Elsharif, Daisy Evans, Andreas Fahlström, Pietro Familiari, Fleur L. Fisher, Bryony Ford, Daniel M. Fountain, Keiko Fox, Francesco Gaillard, Chloé Gelder, Theo Georgious, Shamayitri Ghosh, Aimee Goel, Melissa Gough, Athanasios Grivas, Andrew Gvozdanovic, Allan Hall, Hytham Hamid, Liv Hartrick, Samih Hassan, Jack Henry, Damian Holliman, Abdurrahman I. Islim, Asgeir S. Jakola, Mohsen Javadpour, Michael D. Jenkinson, Sanjeeva Jeyaretna, Adrian Jimenez, Josephine Jung, Andranik Kahramanian, Neeraj Kalra, David O. Kamson, Oliver Kennion, Adham M. Khalafallah, Sarah Kingdon, Ruwanthi Kolamunnage-Dona, Shelli Diane Koszdin, Howra Ktayen, Aditaya Kumar, Simon Lammy, Pierfrancesco Lapolla, Jun Yi Lau, Jing Xian Lee, Ryan Leyden, Patricia Littlechild, Sophie Liu, Darmanin Lora-Kay, Vivia Lung, Stephen T. Magill, Hani J. Marcus, Fawaz E. Marhoom, Javier Martin-Alonso, Ryan K. Mathew, Calan Mathieson, Nathan McSorley, Tobias Mederer, Shaveta Mehta, Torstien R. Meling, Samantha J. Mills, Christopher P. Millward, Andrea Mingoli, Mujtaba Mohammad, Debraj Mukherjee, Amir H. Zamanipoor Najafabadi, Olivia Näslund, Minh Nguyen, Imran Noorani, Gildas Patet, Omar N. Pathmanaban, Andrea Perera, Amit Persad, See Yung Phang, Rory J. Piper, Jonathan Pollock, Benjamin Price, Martin Proescholdt, Richard Pullicino, James Robins, Ola Rominiyi, David

Rowland, Bobby Sachdev, Fozia Saeed, Mahmoud Saleh, Thomas Santarius, Antonio Santoro, Ieva Sataite, Antony Kevin Scafa, Verena Schadewaldt, Syed Wajahat Shah, Mustafa El Sheikh, Zenab Sher, Bente Sandvei Skeie, Alexander Smedley, Agbolahan Sofela, Jerome St George, Torbjørn Strømsnes, Nigel Suttner, William Taylor, Philip Theodosopoulos, Elliot Tilling, Manjul Tripathi, Ismail Ughratdar, James Ulrich, Adithya Varma, Anil Varma, Maria Velicu, Sara Venturini, Esther Wu, Jacob Young, Giuseppa Zancana, Catherine Zhang. International Consortium on Meningioma (ICOM): Karolyn Au, Jill Barnholtz-Sloan, Felix Behling, Linda Bi, Priscilla Brastianos, Chaya Brodie, Nicholas Butowski, Ana Castro, Aaron Cohen-Gadol, Marta Couce, Francesco Dimeco, Katherine J. Drummond, Ian Dunn, Craig Erker, Michelle Felicella, Daniel M. Fountain, Eva Galanis, Norbert Galldiks, Caterina Giannini, Roland Goldbrunner, Oliver Hanemann, Christel Herold-Mende, Luke Hnenny, Craig Horbinski, Raymond Huang, Abdurrahman I. Islim, Mohsen Javadpour, Michael D. Jenkinson, Christine Jungk, Gerhard Jungwirth, Timothy Kaufmann, Boris Krischek, Sylvia Kurz, Daniel Lachance, Christian Lafougere, Katrin Lamszus, Ian Lee, Serge Makarenko, Tathiana Malta, Tiit Illimar Mathiesen, Christian Mawrin, Michael McDermott, Christopher P. Millward, Jennifer Moliterno-Gunel, Andrew Morokoff, Farshad Nassiri, HK Ng, Houtan Noushmehr, Omar N. Pathmanaban, Arie Perry, Laila Poisson, Bianco Pollo, Aditya Ragunathan, David Raleigh, Mirjam Renovanz, Franz Ricklefs, Felix Sahm, Andrea Saladino, Antonio Santacroce, Thomas Santarius, Christian Schichor, Nils Schimdt, Jens Schittenhelm, Warren Selman, Helen Shih, Andrew Sloan, Matija Snuderl, Jim Snyder, Erik Sulman, Suganth Suppiah, Ghazaleh Tabatabai, Marcos Tatagiba, Marco Timmer, Jorg-Christian Tonn, Michael Vogelbaum, Andreas Von Deimling, Tobias Walbert, Simon Walling, Justin Z. Wang, Patrick Wen, Manfred Westphal, Stephen Yip, Gabriel Zada, Gelareh Zadeh, Viktor Zherebitskiy. British Neurosurgical Trainee Research Collaborative (BNTRC): Daniel M. Fountain, Michael T.C. Poon, Angelos Kolias, Julie Woodfield, Rory Piper, Aswin Chari, Abdurrahman I. Islim, Neeraj Kalra.

**Contributors** AII, SJM, ARB and MDJ conceived the study. All drafted the initial study protocol. CPM, RJP, DMF, SM, RK-D, UA, SDK, TG, SJM, ARB, RKM, TS and MDJ provided advice and input on the final protocol. All authors proofread and approved the final manuscript.

**Funding** The authors have not declared a specific grant for this research from any funding agency in the public, commercial or not-for-profit sectors.

**Competing interests** AII is supported by Health Education England (North West) Academic Foundation Programme. CPM is a PhD candidate funded by The Brain Tumour Charity to complete The COSMIC project. DMF is funded by a National Institute for Health Research Academic Clinical Fellowship and Cancer Research UK Predoctoral Bursary. TS founded and leads the Anaplastic Meningioma International Consortium (AMiCo). TS and MDJ cofounded the British-Irish Meningioma Society (BIMS). MDJ received a grant from the National Institute for Health Research Health Technology Assessment programme for the Radiation vs Observation for Atypical Meningioma (ROAM) trial (NIHR ID: 12/173/14). MDJ and SJM received a grant from the National Institute for Health Research Health Technology Assessment programme for Surgeons Trial Of Prophylaxis for Epilepsy in seizure naïve patients with Meningioma (STOP'EM) (NIHR ID: NIHR129748).

**Patient consent for publication** Not applicable.

**Provenance and peer review** Not commissioned; externally peer reviewed.

**ORCID iDs**
Abdurrahman I Islim http://orcid.org/0000-0001-9621-043X
Christopher P Millward http://orcid.org/0000-0001-7727-1157
Ruwanthi Kolamunnage-Dona http://orcid.org/0000-0003-3886-6208
Ryan K Mathew http://orcid.org/0000-0002-2609-9876

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
