## [Reviewer comments · BMJ Open]

ARTICLE DETAILS

TITLE (PROVISIONAL)	External validation and recalibration of an incidental meningioma prognostic model – IMPACT: protocol for an international multicentre retrospective cohort study
AUTHORS	Islim, Abdurrahman; millward, Christopher; Piper, Rory; Fountain, Daniel; Mehta, Shaveta; Kolamunnage-Dona, Ruwanthi; Ali, Usama; Koszdin, Shelli; Georgious, Theo; Mills, Samantha; Brodbelt, Andrew; Mathew, Ryan; Santarius, Thomas; Jenkinson, Michael

VERSION 1 – REVIEW

REVIEWER	Shu, Kai Huazhong University of Science and Technology
REVIEW RETURNED	22-Jul-2021

GENERAL COMMENTS	I have no objection against this manuscript. The study design is scientific and well-organised. I am also looking forward to the results of international multi-center validation of IMPACT model.
--

REVIEWER	Gadelha Figueiredo, Eberval University of Sao Paulo
REVIEW RETURNED	26-Jul-2021

GENERAL COMMENTS	This is a well written paper on a relevant topic.
---

VERSION 1 – AUTHOR RESPONSE

Reviewer: 1

Dr. Kai Shu, Huazhong University of Science and Technology

Comments to the Author:

I have no objection against this manuscript. The study design is scientific and well-organised. I am also looking forward to the results of international multi-center validation of IMPACT model.

Response: Thank you. No changes to address

Reviewer: 2

Dr. Eberval Gadelha Figueiredo, University of Sao Paulo

Comments to the Author:

This is a well written paper on a relevant topic.

Response: Thank you. No changes to address